# Numerical Simulation of Natural Gas Hydrate Exploitation in Complex Structure Wells: Productivity Improvement Analysis

**Hongyu Ye [1], Xuezhen Wu [1,*] and Dayong Li [2]**

[1] College of Civil Engineering, Fuzhou University, Fuzhou 350108, China; 200520015@fzu.edu.cn
[2] School of Storage, Transportation and Construction Engineering, China University of Petroleum, Qingdao 266590, China; ldy@upc.edu.cn
* Correspondence: wu@fzu.edu.cn; Tel.: +86-15059152783

**Abstract:** About 90% of the world's natural gas hydrates (NGH) exist in deep-sea formations, a new energy source with great potential for exploitation. There is distance from the threshold of commercial exploitation based on the single well currently used. The complex structure well is an efficient and advanced drilling technology. The improvement of NGH productivity through various complex structure wells is unclear, and there is no more complete combing. Thus, in order to evaluate their gas production characteristics, we establish a mathematical model for exploitation of NGH, and then 13 sets of numerical models based on the geological parameters of the Nankai Trough in Japan are developed and designed, including a single vertical well, a single horizontal well, 1~4 branch vertical wells, 1~4 branch horizontal wells, and 2~4 branch cluster horizontal wells. The research results indicate that wells with complex structures represented by directional wells and multilateral wells can significantly increase the area of water and gas discharge, especially cluster wells, whose productivity can be increased by up to 2.2 times compared with single wells. Complex structural wells will play an irreplaceable role in the future industrialization of NGH.

**Keywords:** marine natural gas hydrate; complex structural wells; mathematical model; depressurization method; numerical simulation; productivity improvement

## 1. Introduction

Natural gas hydrate (NGH) is widely distributed on the permafrost on the land and seabed of the continental margin. It is considered one of the most potential clean energy sources after coalbed methane, tight gas, and shale gas, with the advantages of wide distribution and large storage capacity. The current methods of the exploitation of NGH reservoirs can be roughly divided into depressurization, thermal stimulation, displacement, and inhibitor injection, as well as their combined application. Many indoor experiments, numerical simulations and field trials have shown that depressurization is the most economical and effective way. In the realized engineering case, from 2002 to 2012, Canada and the United States successively used thermal stimulation, depressurization, and $CO_2$ replacement methods to conduct short-term exploitation of polar sandstone three times [1,2]; from 2013 to 2020, China and Japan used the depressurization method to run 4 test exploitations in the South China Sea and the Nankai Trough in Japan [3–5]. Among them, in 2020, China used horizontal well technology for test production for the first time, achieving continuous natural gas production for 30 days, with a total output of approximately $8.614 \times 10^5$ m³ of natural gas. It made a huge breakthrough, but there is still a certain distance from the threshold of commercial exploitation [6].

From the perspective of increasing capacity, the two basic ways to increase the capacity of a single well are mainly to increase the rate of in situ hydrate decomposition and to expand the areas of hydrate decomposition. For the former, increasing the pressure drop or providing a heat source can achieve the purpose, but in fact, since the existing wellbore is mostly fabricated of the plain concrete structure and the reservoir strength is low, it is

easy to cause wellbore damage or even collapse, while the method of providing a heat source will generate a large amount of heat loss during the process of heat injection, which is less efficient and more costly. Therefore, expanding the hydrate decomposition area may be the best solution to improve the mechanical properties of the wellbore, maintain reservoir stability and increase production capacity. Complex structure wells represented by directional wells (especially horizontal wells) and multi-branch wells can significantly increase the area of water and gas release and will have an irreplaceable role in the future industrialization of NGH.

In recent years, experts and scholars from all over the world have carried out a large number of laboratory tests and simulation studies on the exploitation of NGH complex structure wells. Chong et al. demonstrated, based on small-scale experiments, that horizontal wells help improve continuous gas production cycles and gas recovery by about 5.5% to 10.0% [7]. Mao et al. carried out laboratory simulations of hydrate exploitation from vertical wells and horizontal two-branch wells (90° angle) by using a home-fabricated well simulation experimental setup with complex hydrate structures [8]. Feng et al. compared the recovery capacity of vertical and horizontal wells, and found that horizontal wells can increase NHG production capacity in sandy reservoirs by a factor of 10 [9]. Yu et al. analyzed the production enhancement effect of double horizontal wells, and the results showed that the production capacity of dual horizontal wells was much greater than that of single horizontal wells regardless of the relative spatial position of the two horizontal wells [10]. Li et al. conducted an example study and showed that the main borehole with two inclined holes at a depth of about 60 m can double the gas production capacity while relieving the reservoir outflow [11].

In addition, China's first use of horizontal wells in the Shenhu waters of the South China Sea in 2020 proves that wells with complex structures are of great significance in the future research process of NGH industrialization, and it is not very clear about the improvement of NGH productivity of various complex structure wells, and there is no more complete combing. Therefore, this work evaluates the gas production characteristics of multi-branch vertical wells (MVW), multi-branch horizontal wells (MHW) and cluster horizontal wells (CHW) by numerical simulation, using the Nankai Trough formation in Japan as an example, with the aim of exploring the feasibility of complex structural wells and the status of capacity enhancement. First, a mathematical model for exploitation of NGH was established, which considers the non-isothermal reaction, phase equilibrium process and heat transfer process of the reservoir, and simulates the exploitation of NGH by adding hydrate formation and decomposition reaction kinetic equations. Then, a vertical well (VW) model for the first NGH exploitation in the Nankai Trough in Japan was developed and fitted to the production characteristics curve of the actual project to verify the feasibility of the model. Finally, a complex structured well including a single horizontal well (HW), 1~4 branch vertical wells (MVW1~4), 1~4 branch horizontal wells (MHW1~4), and 2~4 branch cluster horizontal wells (CHW2~4), were designed based on this geological model, and the depressurization method simulation evaluated its gas production characteristics.

## 2. Model Establishment and Verification

### 2.1. Mathematical Model Establishment

For the numerical analysis of NGH production capacity prediction, a series of mature hydrate exploitation simulators were developed internationally, such as TOUGH + HYDRATE, MH-21HYDRES and CMG-STARS [12–14]. In this work, the CMG-STARS code was used to simulate the exploitation of NGH reservoirs by adding hydrate formation and decomposition reaction kinetic equations [15,16].

Consider the following problem statement: (i) Consider an NGH reservoir with a porous structure. The pores of the reservoir are saturated with methane and methane hydrate. (ii) Consider three phases (gas phase, water phase, solid phase) and four components (free gas component, decomposition gas component, water component, and hydrate

component). Among them, the gas phase contains only methane gas, and the hydrate is treated as a solid phase; (iii) only consider the two-phase flow of gas and liquid, and the fluid seepage conforms to Darcy's law; (iv) homogeneous formation, i.e., porosity, permeability is constant; (v) neglect the diffusion of gas and the dissolution of gas in water [17,18].

According to the above problem statement, the mass conservation equation of each component is as follows:

Free/decomposition gas component:

$$\nabla\left(\frac{\rho_g k k_{rg}}{\mu_g}\nabla\rho_g\right) + q_g + \dot{m}_g = \frac{\partial}{\partial t}\left(\rho_g \phi S_g\right) \tag{1}$$

Water component:

$$\nabla\left(\frac{\rho_w k k_{rw}}{\mu_w}\nabla\rho_w\right) + q_w + \dot{m}_w = \frac{\partial}{\partial t}\left(\rho_w \phi S_w\right) \tag{2}$$

Hydrate component:

$$\frac{\partial}{\partial t}\left(\rho_h \phi S_h\right) = \dot{m}_w + \dot{m}_g \tag{3}$$

where $\rho_i$ is the density of each phase, kg/m$^3$($i = g$, $w$, $h$); $k$ is permeability, mD; $k_{ri}$ is relative permeability of each phase, mD; $\mu_i$ is viscosity of each component, mPa s; $q_i$ is injection/output quality per unit time and unit volume, kg/m$^3$/s; $S_g$, $S_w$, and $S_h$ are, respectively, the saturation of gas, water and hydrate; $\dot{m}_i$ is the masses of gas, water, and hydrates decomposed per unit time, kg/s; $\phi$ is porosity of the medium.

The energy conservation equation in a unit volume of porous media is:

$$\frac{\partial}{\partial t}\left(C_{eff}T\right) = \nabla\left(\lambda_{eff}\nabla T\right) + \nabla\left[\left(\frac{k k_{rw}\rho_w}{\mu_w}C_w\nabla\rho_w + \frac{k k_{rg}\rho_g}{\mu_g}C_g\nabla\rho_g\right)T\right] - m_h\Delta H_h + m_i\Delta H_i + \left(q_w C_w + q_g C\right)_g T \tag{4}$$

here $C_{eff}$ and $\lambda_{eff}$ are:

$$C_{eff} = (1-\phi)\rho_r C_r + \phi S_g \rho_g C_g + \phi S_w \rho_w C_w + \phi S_h \rho_h C_h + \phi S_i \rho_i C_i \tag{5}$$

$$\lambda_{eff} = (1-\phi)\lambda_r + \phi S_g \lambda_g + \phi S_w \lambda_w + \phi S_h \lambda_h + \phi S_i \lambda_i \tag{6}$$

where $\rho_r$ is the density of rock, kg/m$^3$;$C_r$, $C_g$, $C_w$, $C_h$, and $C_i$ are, respectively, the specific heat of rock, gas, water, hydrate and ice, J/g/K. $\lambda_r$, $\lambda_g$, $\lambda_w$, $\lambda_h$, and $\lambda_i$ are, respectively, the thermal conductivity of rock, gas, water, hydrate and ice, W/m/K. $\Delta H_h$, $\Delta H_i$ are the heat absorbed/released during the decomposition of hydrate and ice per mole, J/mol; $S_i$ is saturation of ice phase.

The basic hydrate dissociation equation is given by:

$$CH_4 \cdot N_h H_2O_{(solid)} \leftrightarrow CH_{4(gas)} + N_h H_2O_{(liquid/ice)} \tag{7}$$

where $N_h$ is the hydration number.

In CMG-STARS, the equations below where used to calculate the decomposition and formation of NGH:

$$\frac{dc_h}{dt}_{decay} = A\exp\left(\frac{-\Delta E}{RT}\right)(\phi S_w \rho_w)(\phi S_h \rho_h y_i p_g)\left(1 - \frac{1}{K(P,T)}\right) \tag{8}$$

$$\frac{dc_h}{dt}_{form} = B(1+\phi S_h)\exp\left(\frac{-E}{RT}\right)(\phi S_w \rho_w)\left(\frac{1}{K(P,T)} - 1\right) \tag{9}$$

here $A$ and $B$ are the kinetic rate constants of hydrate decomposition and formation:

$$A = \frac{k_d^o A_{hs}}{\rho_w \rho_h} \;\& \; B = \frac{k_f^o A_{hs}}{\rho_w} \tag{10}$$

where (12) represents the decomposition of hydrate, and (13) represents the formation of hydrate; $c_h$ is the hydrate concentration, gmol/m$^3$; $t$ is time, s; $A_{hs}$ is the decomposition surface area, m$^2$; $P_g$ is the gas phase pressure, kPa; $E$ is the activation energy, J; $y_i$ is the mole fractions of methane in gas and liquid phase; $k_f^o$ is the intrinsic formation rate constant; $K$ is a function of pressure ($P$) and temperature ($T$), describing the equilibrium state of hydrate:

$$K = \frac{k_1}{P} \exp\left(\frac{k_2}{T - k_3}\right) \tag{11}$$

where $k_1$, $k_2$ and $k_3$ are the fitting parameters, which can be generated either in the form of coefficients or in the form of tables (Figure 1).

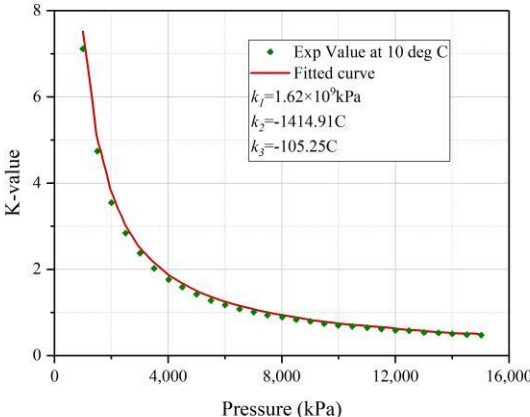

**Figure 1.** K-value curve under NGH three-phase balance.

The above only lists some of the governing equations and differential equations of the model. The specific NGH decomposition/formation principle, numerical method and the derivation of differential Equations (10) and (11) are shown in Appendix A.

### 2.2. Reservoir Characteristics

In March 2013, the world's first sea hydrate exploitation test was completed in the Nankai Trough in Eastern Japan [3]. The trial exploitation lasted for 6 days, but it was finally terminated due to a large amount of sand production. A total of four wells were drilled at station AT1 on the trial exploitation site, of which AT1-P was used as a depressurization exploitation well [19]. The results of the geophysical logging and sampling analysis showed that the hydrate layer at the AT1 site is about 60 m thick, and it mainly exists in unconsolidated sand-rich sediments, and the sand layer forms an interlayer structure with a clay or silt layer above and below [20]. The stratum is generalized as a horizontally extended stratum without considering the influence of the dip angle of the stratum on the flow of gas and water in both phases, and the upper and lower sides of the model are set as constant temperature and constant pressure boundaries where fluid migration and heat exchange can occur:

$$\begin{cases} P(x,y,z,t)|_{(x,y,z)\in\Gamma} = P_e(x,y,z,t) \\ \left.\frac{\partial P}{\partial n}\right|_{(x,y,z)\in\Gamma} = 0 \end{cases} \tag{12}$$

$$\begin{cases} T(x,y,z,t)|_{(x,y,z)\in\Gamma} = T_0(x,y,z,t) \\ \left.-C\frac{\partial T}{\partial n}\right|_{(x,y,z)\in\Gamma} = q \end{cases} \tag{13}$$

where $n$ is the number of NGH; $C$ is the index (the value in this paper is 1.57); $q$ is a constant.

The initial formation pressure field distribution of the model in this paper is determined by the change in the seafloor surface pressure of 10.72 MPa according to the pressure gradient of 10 kPa/m, and the temperature field distribution is determined by the seafloor surface temperature of 3.75 °C according to the change in the geothermal gradient of 0.03 °C/m. Then a three-dimensional geological model is established (Figure 2).

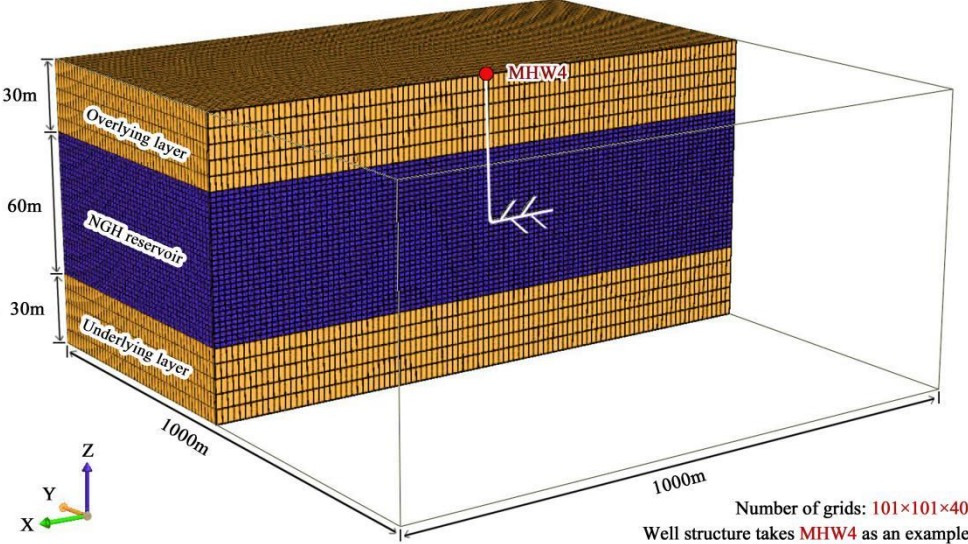

**Figure 2.** Geological model of NGH reservoir by depressurization.

To minimize the influence of boundary conditions, the model is expanded to 1000 m in X and Y directions; the total thickness in Z direction is 90 m, of which the total thickness of the NGH reservoir is 60 m; the thickness of overburden layer is 30 m; the thickness of the underlying layer is 30 m. The grids of the overlying layer and the underlying layer are argillaceous sediments (permeability less than 15 mD) and sandy sediments (permeability greater than 15 mD), with a size of 10 m × 10 m × 6 m. The grid size of the NGH reservoir is 10 m × 10 m × 2 m, and the specific reservoir parameters and other stratigraphic parameters are shown in Table 1. A total of 408,040 grids (101 grids along the x, y-coordinate, and 40 grids along the z-coordinate) is used for the numerical simulations in this study.

**Table 1.** Numerical simulation model parameters for trial exploitation in Nankai Trough, Japan.

| Parameter | Value | Parameter | Value |
|---|---|---|---|
| NGH reservoir thickness/(m) | 60 | NGH saturation | 0.6 |
| Thickness of overlying/underlying layer/(m) | 30 | Porosity | 0.4 |
| Geothermal gradient/(°C/m) | 0.03 | Lateral permeability/(mD) | 10 |
| Longitudinal permeability/(mD) | 5 | Pressure gradient/(kPa/m) | 10 |
| NGH molar mass/(Kg/gmole) | 0.119543 | Top initial pressure/(MPa) | 13.5 |
| Top initial temperature/(°C) | 12.1 | NGH density/(Kg/m$^3$) | 919.7 |
| Thermal conductivity of rock/(W/m/K) | 1.73 | NGH thermal conductivity/(W/m/K) | 0.5 |
| Thermal conductivity of water/(W/m/K) | 0.69 | Bottom hole production pressure/(MPa) | 4.5 |
| Gas composition | 100%CH$_4$ | Heat of decomposition of NGH/(J/mole) | 51,858 |

### 2.3. Wellbore Structure Grouping

Since the 1980s, the emergence of guided drilling technology and branch completion technology led to revolutionary progress in complex structure wells, with increasing application scale and broad application areas. The complex structure well refers to all wells with a more complex structure or process except conventional straight wells, including

horizontal, radial horizontal, lateral drilling horizontal, large displacement and many other drilling wells, which are expected to enhance the production capacity further and gradually realize industrial exploitation if it can be applied to offshore NGH exploitation. Currently, horizontal wells or complex structured wells with horizontal wells as the primary feature are considered the advanced technology for the efficient development of NGH. The corresponding drilling and exploitation engineering technology are highly challenging, but significant progress has been made in related research [21]. The wellbore established in this paper is mainly composed of four groups (Figure 3). In this case, (i) The control fitting group consists of a vertical well and a horizontal well, the vertical well is consistent with the parameters of the first decompression exploitation in the Nankai Trough and is used to verify the accuracy of the model. The horizontal well main length (L) = 100 m and both serve as reference objects. (ii) The main borehole length (L) = 38 m of the multi-branch vertical well set is composed of 1 to 4 branch wells, which are located at 5 equal points of the main well end, and their length is L = 30 m, and the angle with the main well section and other adjacent branches are 90°. (iii) The length of the main borehole of the multi-branch horizontal well group is L = 100 m, which is composed of 1 to 4 branch wells, and the branch wells are located at 5 equal divisions at the end of the main well. Their length is L = 51.3 m, and the angle between them and the main shaft section is 45°. (iv) The cluster horizontal well group consists of 2 to 4 horizontal wells with length (L) = 100 m and spacing (d) = 100 m, all at a uniform height. The above wellbore structures are all located in the central area of the geological model, and the radius $r_b$ of the wellbore is all set to 0.1 m, and the exploitation pressure is set to 4.5 MPa.

### 2.4. Verification of the Feasibility of Model

The exploitation cycle in 2013 lasted for 6 days, with the average daily gas production of approximately $2 \times 10^4$ m$^3$. In the exploitation test, the production well ($r_b$ = 0.1 m) was located in the center of the model and the 38 m shot-hole section was located at the top of the methane hydrate enrichment zone (MHEZ). As in the site test-hole section design scenario, it is planned to use the hydrate-bearing formation (low effective permeability) in the lower part of the MHEZ to block any significant influx of free water from the underlying sandy aquifer into the production well.

Figure 4a shows the comparison between the gas production rate ($Q_t$) of the numerical model and at the site. By comparing both of them with the time evolution characteristics, the field trial production process can be divided into two phases: (i) the wellbore impact phase; and (ii) the initial reservoir gas production capacity phase. At the beginning of the first phase (about the first 3 days), the water in the well was pumped out, followed by a gradual decrease in the bottom pressure of the exploitation well to 4.5 MPa, resulting in a significant drop in the surrounding formation pressure, which led to rapid decomposition of the hydrate around the well and a sharp increase in the $Q_t$ at the wellhead. In the second stage (after the 3rd day of exploitation), the mud and sand gradually blocked the pore channel in the actual production, and the bottomhole pressure was no longer sufficient to further decompose the hydrate, so the actual $Q_t$ began to be lower than the simulated $Q_t$, and the model predicted $Q_t$ reached a maximum of 20,465 m$^3$/d during the 6-day exploitation cycle, which was very close to the actual $Q_t$. Although there are differences in the fluctuation degree between the two, the average values are basically the same. Figure 4b shows the comparison between the numerical model predicted water production rate ($Q_w$) and the actual measured $Q_w$ at the site. As the reservoir is continuously recharged by water sources and the sand out phenomenon causes a large amount of water out at the site, the numerical model predicted water production curve is overall lower than the actual water production rate curve, in addition to the same fluctuation difference between the simulated water production curve and the actual water production curve and the $Q_t$ curve. In summary, it can be considered that the model can better simulate the on-site exploitation situation and has high reliability.

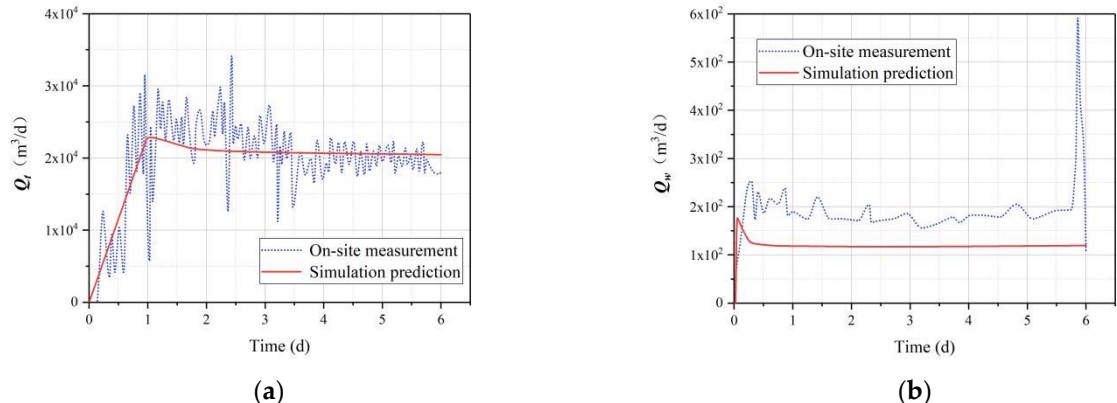

**Figure 3.** Complex wellbore structure grouping.

**Figure 4.** Comparison of simulation results in Nankai Trough, Japan: (**a**) gas production; (**b**) water production.

## 3. Simulation Results of Depressurization

### *3.1. Single Vertical Well and Single Horizontal Well*

Figure 5 shows, for 1 year, the gas production characteristic curves of VW and HW. The $Q_t$ of HW was greater than that of VW [22], and the cumulative gas production ($V_c$) in the first 6 days was about 2.4 times that of VW, and the increase in production was about 1.5 times from the $V_c$ in one year. It confirms that the horizontal well increases the contact area between the wellbore and the NGH reservoir compared with the vertical well, which expands the NGH decomposition front and multiplies the amount of NGH involved in the decomposition at the same time [23]. Further research by Feng et al. found that under the condition that the contact surface of horizontal wells and vertical wells are close to the reservoir, the recovery rate of the reservoir temperature in the later stage of horizontal well production is greater than that of vertical wells [24]. Admittedly, horizontal wells are more conducive to NGH exploitation than vertical wells, but relying solely on pressure reduction methods combined with HW is not sufficient for commercial exploitation.

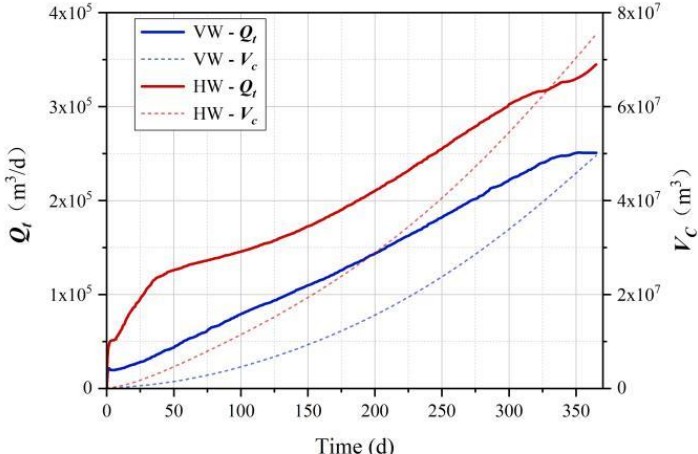

**Figure 5.** Gas production characteristic curve of VW and HW.

### *3.2. Multi-Branch Vertical Well*

Figure 6 shows the gas production characteristic curves of MVW. In addition to $Q_t$, the NGH exploitation is controlled by the drop in pressure, the contact area also has a huge impact. When using a vertical well as the main structure for NGH exploitation, $Q_t$ shows an increase followed by a decrease regardless of the branch well arrangement used for depressurization. This is due to the presence of more free gas around the wellbore in the early stage of depressurization due to other factors such as disturbances in the drilling process. From the pressures measured ($P_w$) at the monitoring points, the greater the number of branch wells, the faster the $P_w$ at the main well location reaches the expected 4.5 MPa. With essentially the same $P_w$ and pressure drop, each additional branch well can achieve a production increase of about 20%. In the late stage of exploitation, the $Q_t$ curve of VW tends to flatten out due to the limitation of the contact area, while the $Q_t$ curve of MVW still increases.

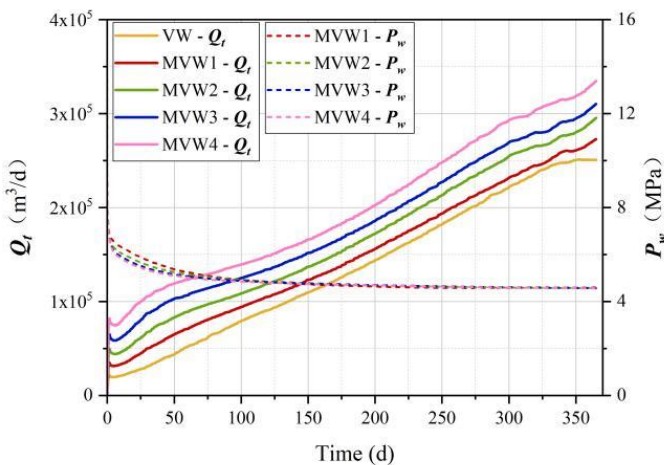

**Figure 6.** Gas production characteristic curve of MVW.

*3.3. Multi-Branch Horizontal Well*

The production increase effect of MHW is not as significant as MVW (Figure 7). The diagonal branch well arrangements used in this paper are all in one plane and have short spacing, resulting in a similar trend in the mid- to late-stage curves for MHW1-3, except for a higher $Q_t$ lift compared with HW at the beginning of exploitation (first 30 days), and almost coincides with the $P_w$ curves as well. However, when the number of diagonal branch wells reaches four, the $Q_t$ of MHW4 is more significantly improved than HW, and the combined production increase is about 5%. The following factors may cause the poor yield increase: (i) The contact area added by the horizontal main shaft is already large enough. In field projects or simulation tests, the horizontal main shaft length of MHW is usually over 100 m, while the vertical main shaft of MVW is generally in several tens of meters. (ii) Branch wells are typically operated in lengths of several tens of meters, and unlike vertical main wells, their increased contact area has little effect compared with horizontal main wells and produces a high degree of overlap in the range of pressure reduction. Therefore, it is essential to choose a reasonable layout plan for complex structural wells while weighing the cost and the effect of increasing production, because a single-minded increase in the depressurization amplitude or contact area may be counterproductive.

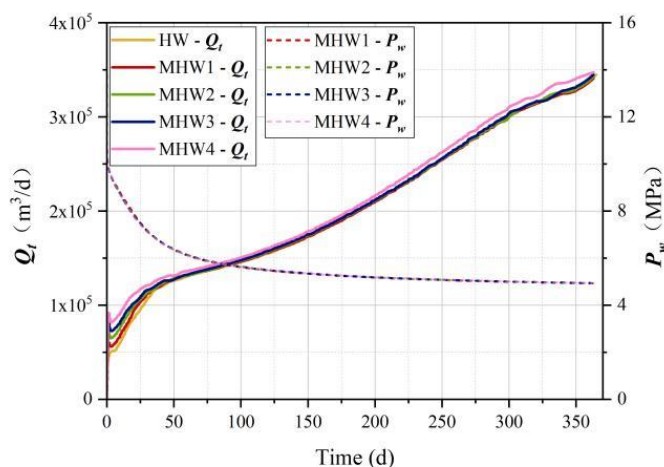

**Figure 7.** Gas production characteristic curve of MHW.

*3.4. Cluster Horizontal Well*

The effect of increasing the production of CHW was very satisfactory (Figure 8). Due to the distribution characteristics of CHW, the contact area of the wellbore and the NGH reservoir was more extensive, so higher natural gas production can be obtained. Especially

in the early stage, the production volume was significantly increased compared with HW; the early-stage peaks of CHW2~4 were 1.8 times, 2.8 times and 3.7 times that of HW, respectively, with a combined capacity ramp-up of approximately 44%, 81% and 120%, respectively, in one year. Since the pressure measurement point was connected to one of the horizontal sub-wells in the layout of CHW3, the rate of pressure drop was faster than that of CHW2 and CHW4. Compared with the existing well pattern exploitation theory [10,25,26], due to differences in structure and layout, cluster horizontal wells do not belong to well pattern exploitation in a strict sense. However, CHW can also have the synergistic effect of the well pattern, which can significantly increase the production capacity of NGH. In addition, establishing a large number of numerical models and laboratory tests to analyze the parameters such as the length, spacing and direction of the CHW split wells are of great significance to the realization of commercial NGH exploitation and is also a future research direction.

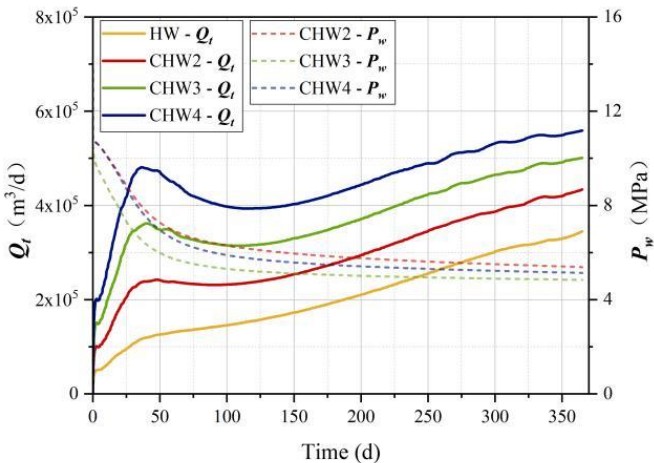

**Figure 8.** Gas production characteristic curve of CHW.

## 4. Discussion

The daily production rate of $5 \times 10^5$ m$^3$/d adopted by many scholars, is taken as the threshold value, and the average daily gas production value ($Q_d$) of this numerical simulation are compared with this threshold value (Figure 9). The technical threshold and operating cost of VW are low, and it is the main well structure for NGH test production. However, VW has a low single well capacity, averaging tens to hundreds of thousands of cubic meters of gas per day and a small gas production range. Moreover, the life span of VW is expected to be short, with the longest-lasting only 60 days from the completed marine exploitation cases. HW can substantially increase the contact area with NGH, and the longer the length of HW, the larger the decomposition surface, which has a greater increase in productivity. In addition, compared with a single well, MVW, MHW and CHW can all increase certain productivity, of which CHW3 is only 10% away from the threshold. Overall, the present simulation results verify that the well type has significant control over the hydrate exploitation capacity. When the pressure drop is the same, the more hydrates that are decomposed when the NGH is produced under a pressure drop in complex structure wells, the better the gas production and the longer the gas production time. In particular, CHW can significantly improve production efficiency, and if a more substantial step-down can be achieved, it is very likely to break the threshold [27,28].

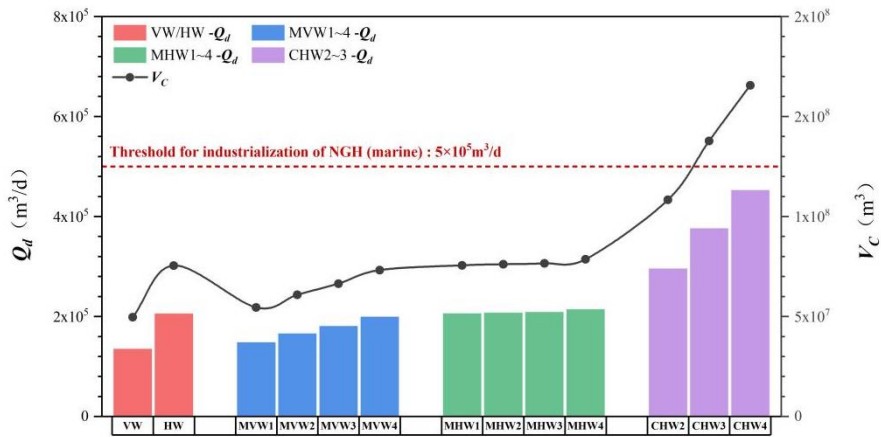

**Figure 9.** Comparison chart of capacity improvement summary.

Nevertheless, whether it is land or marine NGH drilling, due to conventional drilling methods, bottom hole pressure fluctuations, NGH decomposition around the wellbore, and other factors, the instability of the wellbore and surrounding formations are aggravated. At the same time, offshore drilling also faces challenges in terms of water depth, seabed temperature, ocean currents, and seabed subsidence. Therefore, for drilling and production in the sea area of complex wells, pressure fluctuations in NGH reservoirs should be avoided as much as possible, effective control of temperature-pressure in the wellbore should be maintained, and the balanced installation of liners, screens and completion facilities should be maintained. Because of the above difficulties, a series of further studies are needed in the future: (i) Explore more realistic mathematical models of NGH decomposition, reservoir mechanical characteristics and constitutive relationships to ensure long-term safe and stable development. (ii) Strengthen the research of drilling technology, break through the technical difficulties such as the difficulty of stabilizing the tool face of shallow deflection and the difficulty of meeting the requirement of deflection rate, and actively explore the drilling and completion technology of complex structure wells such as multilateral wells or cluster wells in shallow soft formations. (iii) Research and development of supporting marine engineering technologies. Small drilling vessels with coiled casing or a combination of submersible drilling equipment should be improved and developed. Technologies such as pressure-controlled drilling, underbalanced drilling, casing drilling, and insulated vertical tubing drilling should be fully applied to solve the problems such as a wellhead collapse, well wall instability, and a subsea landslide, which may be caused during NGH drilling.

## 5. Conclusions

In this work, a total of 13 sets of numerical models, including a single vertical well, single horizontal well, 1~4 branch vertical wells, 1~4 branch horizontal wells and 2~4 branch cluster wells, were established and designed based on the parameters related to the first Japanese decompression exploitation in the Nankai Trough to explore the capacity enhancement of the complex structure wells for marine NGH, and the following conclusions were drawn:

a. A mathematical model of NGH reservoir exploitation was established, taking into account the phase equilibrium of hydrate decomposition, hydrate decomposition kinetics, mass conservation, energy conservation, heat conduction and heat convection.

b. Using CMG-STARS, a total of 13 sets of numerical models of complex structure wells were established, and then the reliability of the model was verified by adopting the first test production parameters of the Nankai Trough in Japan and fitting them with the gas and water production data.

c. The simulation results show that, when the pressure drop is the same, complex structure wells can increase the contact area compared with a single well, increas-

ing productivity. In particular, CHW4 has the most significant improvement in exploitation efficiency, 2.2 times that of HW in a one-year exploitation cycle.

d.　Complex structure wells have certain application prospects in the exploitation of marine NGH, but they also face a series of wellbore and formation instability problems. In the future, it is necessary to further improve the existing numerical models, strengthen the research of drilling technology and technology, and develop related marine engineering supporting facilities to ensure safe and stable exploitation.

**Author Contributions:** Conceptualization, H.Y. and X.W.; methodology, X.W.; software, D.L.; validation, X.W.; formal analysis, H.Y.; investigation, H.Y.; resources, X.W.; data curation, H.Y.; writing—original draft preparation, H.Y.; writing—review and editing, X.W.; visualization, H.Y.; supervision, X.W.; project administration, X.W.; funding acquisition, X.W. All authors have read and agreed to the published version of the manuscript.

**Funding:** This research was funded by the National Natural Science Foundation of China (No. 41907251, 52179098), Natural Science Foundation of Fujian Province (No.2019J05030) and the Major Basic Research Project of Shandong Provincial Natural Science Foundation (No. ZR2019ZD14).

**Institutional Review Board Statement:** Not applicable.

**Informed Consent Statement:** Not applicable.

**Data Availability Statement:** Not applicable.

**Conflicts of Interest:** The authors declare no conflict of interest.

**Abbreviations**

The following abbreviations are used in this manuscript:

| | |
|---|---|
| NGH | Natural gas hydrate |
| MHEZ | Methane hydrate enrichment zone |
| VW | Vertical well |
| HW | Horizontal well |
| MVW1~4 | Multi-branch vertical well (1~4 Branch) |
| MHW1~4 | Multi-branch horizontal well (1~4 Branch) |
| CHW2~4 | Cluster horizontal well (2~4 Branch) |

**Nomenclatures**

The following nomenclatures are used in this manuscript:

| | |
|---|---|
| $A$, $B$, $C$, $q$ | constant |
| $A_d$ | total surface area of the hydrate particles (m$^2$) |
| $A_{hs}$, $A_{dec}$ | decomposition surface area (m$^2$) |
| $c_h$ | hydrate concentration (gmol/m$^3$) |
| $C_r$, $C_g$, $C_w$, $C_h$, $C_i$ | specific heat of rock, gas, water, hydrate and ice (J/g/K) |
| $E$ | the activation energy (J) |
| $f_e$ | fugacity of methane at the three phase equilibrium condition |
| $f_g$ | fugacity of methane in the gas phase |
| $k$ | permeability (mD) |
| $k_d$ | hydrate decomposition rate constant |
| $k^o_d$ | intrinsic decomposition rate constant |
| $k^o_f$ | intrinsic formation rate constant |
| $k_{rg}$, $k_{rw}$, $k_{rh}$ | relative permeability of each phase (mD) |
| $\dot{m}_g$, $\dot{m}_h$, $\dot{m}_w$ | masses of gas, water, and hydrates decomposed per unit time (kg/s) |
| $n$, $N_h$ | hydration number |
| $P_e$ | hydrate three-phase equilibrium pressure (kPa) |
| $P_g$ | gas phase pressure (kPa) |
| $P_w$ | pressures measured at the monitoring points (MPa) |
| $q_g$, $q_w$, $q_h$ | injection/output quality per unit time and unit volume (kg/m$^3$/s) |

| | |
|---|---|
| $Q_d$ | average daily gas production value (m$^3$/d) |
| $Q_t$ | gas production rate (m$^3$/d) |
| $Q_w$ | water production rate (m$^3$/d) |
| $R$ | universal gas constant |
| $S_g, S_w, S_h, S_i$ | saturation of gas, water, hydrates and ice |
| $t$ | time (s) |
| $v$ | reaction speed (m$^3$/s) |
| $Vc$ | cumulative gas production (m$^3$) |
| $x_i, y_i$ | mole fractions of methane in gas and liquid phase |
| $\Delta H_h, \Delta H_i$ | heat absorbed/released per mole (J/mol) |
| $\lambda_r, \lambda_g, \lambda_w, \lambda_h, \lambda_i$ | thermal conductivity of rock, gas, water, hydrate and ice (W/m/K) |
| $\mu_g, \mu_w, \mu_h$ | viscosity of each component (mPa s) |
| $\rho_g, \rho_w, \rho_h, \rho_r$ | density of gas, water and hydrates and rock (kg/m$^3$) |
| $\phi$ | porosity of the medium |

## Appendix A. The Specific NGH Decomposition/Formation Principle, Numerical Method and the Derivation of Differential Equation

The essential principle of NGH decomposition or formation is to use specific physical and chemical means to decompose the in-situ natural gas hydrate into gas-water in two phases (Figure A1).

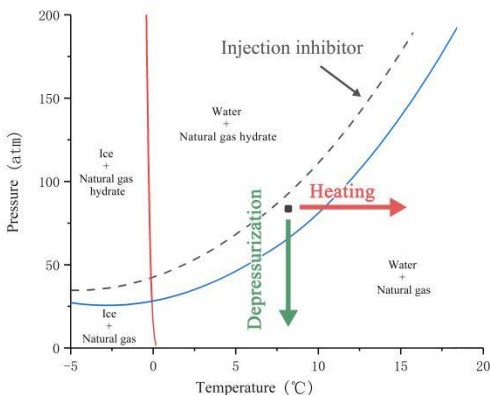

**Figure A1.** Phase diagram of NGH.

Thus, For the presented problem statement for the phase equilibrium differential equation, the basic hydrate dissociation equation is given by:

$$CH_4 \cdot N_h H_2 O_{(solid)} \leftrightarrow CH_{4(gas)} + N_h H_2 O_{(liquid/ice)} \tag{A1}$$

In 1987, Kim et al. conducted the first quantitative study of NGH decomposition kinetics and proposed the following NGH decomposition rate equation [29]:

$$\frac{dc_h}{dt} = k_d A_d (P_e - P_g) \tag{A2}$$

where $c_h$ is the hydrate concentration, gmol/m$^3$; $t$ is time, s; $k_d$ is the hydrate decomposition rate constant; $A_d$ is the total surface area of the hydrate particles, m$^2$; $P_e$ is the hydrate three-phase equilibrium pressure, kPa; $P_g$ is the gas phase pressure, kPa.

The decomposition rate constant $k_d$ is defined as follows:

$$k_d = k_d^o \exp\left(-\frac{E}{RT}\right) \tag{A3}$$

where $k^o_d$ is the intrinsic decomposition rate constant; $E$ is the activation energy, J; $R$ is the universal gas constant.

For the dissociation of hydrate, assuming that $c(t)$ is the concentration of NGH at a certain moment, the following reaction must be simulated:

$$\frac{dc(t)}{dt} = k^0_d f(P, T, A_{dec})$$

(A4)

In CMG-STARS, the reaction speed $v$ is as follows:

$$v = \frac{dc(t)}{dt} = k^0_d A_{dec} (f_e - f_g) \exp\left(-\frac{\Delta E}{RT}\right)$$

(A5)

where $f_e$ is the fugacity of methane at the three-phase equilibrium condition; $f_g$ is the fugacity of methane in the gas phase. Considering that the fugacity coefficient is equal to 1.0, the fugacity can be approximated as the equivalent pressure:

$$v = \frac{dc(t)}{dt} = k^0_d A_{dec} (P_e - P_g) \exp\left(-\frac{\Delta E}{RT}\right) = k^0_d A_{dec} P_e \left(1 - \frac{P_g}{P_e}\right) \exp\left(-\frac{\Delta E}{RT}\right)$$

(A6)

where, the ratio of $Pe/Pg$ can be considered as partial equilibrium $K$-value. This $K$-value can be obtained from the laboratory three-phase equilibrium data (Figure 1):

$$K = \frac{k_1}{P} \exp\left(\frac{k_2}{T - k_3}\right)$$

(A7)

where $A_{dec}$ is the decomposition surface area, m$^2$. Generally, hydrates exist in porous media and are composed of spherical particles with a surface area of $A_{hs}$. Then the effective decomposition area of hydrates per unit volume of porous media can be approximated as:

$$A_{dec} = \phi^2 A_{hs} S_w S_h$$

(A8)

Including Equation (A2) into Equation (A8), and let $c_h = c(t)$:

$$\frac{dc_h}{dt} = k^0_d \phi^2 A_{hs} S_w S_h p_e \left(1 - \frac{1}{K(P,T)}\right) \exp\left(\frac{-\Delta E}{RT}\right)$$

(A9)

$$\frac{dc_h}{dt} = \left(\frac{k^0_d A_{hs}}{\rho_w \rho_h}\right) (\phi S_w \rho_w)(\phi S_h \rho_h) p_e \left(1 - \frac{1}{K(P,T)}\right) \exp\left(\frac{-\Delta E}{RT}\right)$$

(A10)

By denoting the partial pressure in the gas phase as $y_i P_g$, then based on the Raoult's Law, the equilibrium pressure can be defined as:

$$P_e = \frac{y_i P_g}{x_i}$$

(A11)

where, $y_i$ and $x_i$ are mole fractions of methane in the gas and liquid phase. It is assumed that $x_i = 1$ in the three-phase system of liquid water, hydrate and vapor. Inserting Equation (A11) into (A10) we obtain the final equation:

$$\frac{dc_h}{dt}_{decay} = \left(\frac{k^0_d A_{hs}}{\rho_w \rho_h}\right) (\phi S_w \rho_w)(\phi S_h \rho_h)(y_i p_g) \left(1 - \frac{1}{K(P,T)}\right) \exp\left(\frac{-\Delta E}{RT}\right)$$

(A12)

The same can be obtained:

$$\frac{dc_h}{dt}_{form} = \left(\frac{k^o_f A_{hs}}{\rho_w}\right) (1 + \phi S_h)(\phi S_w \rho_w) \left(\frac{1}{K(P,T)} - 1\right) \exp\left(\frac{-E}{RT}\right)$$

(A13)

The above Equations (10) or (A12) and (11) or (A13) are the formation or decomposition models of NGH in CMG-STARS. In addition, the mathematical model of NGH mining should also include the mass conservation equations and energy conservation equations of each component in the text. The calculation and solution process is mainly as follows: (i) enter the size of the simulation area, grid division, stratum parameters, boundary conditions and initial conditions; (ii) import complex structure well point parameters and work arrangement; (iii) calculate the decomposition rate of NGH; (iv) calculate the saturation of each phase and the fraction of gas phase components; (v) update the model parameters after each analysis step, and judge the control conditions for the end of the calculation, if the end time is not reached, then skip to process (ii) and continue the calculation.

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
