# Peer review of "Numerical Simulation of Natural Gas Hydrate Exploitation in Complex Structure Wells: Productivity Improvement Analysis"

_mathematics, doi:10.3390/math9182184_

Round 1

Reviewer 1 Report

The manuscript shows a numerical simulation case study with a model developed by company Computer Modeling Group Ltd. In current form it is more suitable for journals with main focus on sustainability and/or energy topics (e.g. MDPI journals Energies or Sustainability).

There is no information on discretization of PDE model defined in the manuscript neither on numerical method used for solving it. There are just geometry and number of numerical cells (mesh information), but no information on meaning of each cell. Methodology also does not provide information on solution technique used.

The manuscript must show more information regarding not only above-mentioned comments, but also problem statement (BCs, ICs, etc.) before it can be accepted as a full paper published in the journal Mathematics.

Author Response

Response to Reviewer 1 Comments

Point 1: The manuscript shows a numerical simulation case study with a model developed by company Computer Modeling Group Ltd. In current form it is more suitable for journals with main focus on sustainability and/or energy topics (e.g. MDPI journals Energies or Sustainability).

Response 1:

Thanks for your suggestion. MDPI journals Energies or Sustainability are also excellent journals, and they are also our long-term focus. The good news is that there is special issue “Numerical Simulation and Control in Energy Systems problems” in the engineering mathematics section of Mathematics journals. The authors believe that this paper is particularly suitable for the special issue.

Point 2: There is no information on discretization of PDE model defined in the manuscript neither on numerical method used for solving it. There are just geometry and number of numerical cells (mesh information), but no information on meaning of each cell. Methodology also does not provide information on solution technique used.

Response 2: 

Thanks for your comment

  1. For the explanation of the equation, in order to facilitate understanding and reading, we simplified the description of the formula in Section 2.1. At the same time, appended Appendix A at the end of the article to explain the derivation and establishment of the equation.
  2. For thenumerical method and methodology, the following content has been added to Appendix A:

“The calculation and solution process is mainly as follows: (i) Enter the size of the simulation area, grid division, stratum parameters, boundary conditions and initial conditions; (ii) Import complex structure well point parameters and work arrangement; (iii) Calculate the decomposition rate of NGH; (iv) Calculate the saturation of each phase and the fraction of gas phase components; (v) Update the model parameters after each analysis step, and judge the control conditions for the end of the calculation. If the end time is not reached, then skip to process (ii) and continue the calculation.”

  1. For cell grid information, the following content has been modified in section 2.2:

“To minimize the influence of boundary conditions, the model is expanded to 1000 m in X and Y directions; the total thickness in Z direction is 90 m, of which the total thickness of the NGH reservoir is 60 m; the thickness of overburden layer is 30 m; the thickness of the underlying layer is 30 m. The grids of the overlying layer and the underlying layer are argillaceous sediments (permeability less than 15mD) and sandy sediments (permeability greater than 15mD), with a size of 10m×10m×6m. The grid size of the NGH reservoir is 10m×10m×2m, and The specific reservoir parameters and other stratigraphic parameters are shown in Table 1. A total of 408040 grids (101 grids along the x, y-coordinate, and 40 grids along the z-coordinate) is used for the numerical simulations in this study.”

Point 3: The manuscript must show more information regarding not only above-mentioned comments, but also problem statement (BCs, ICs, etc.) before it can be accepted as a full paper published in the journal Mathematics.

Response 3:

Thanks for your comment. We have completed and revised the problem statement of this model in Section 2.1:

“Consider the following problem statement. (i) Consider an NGH reservoir with a porous structure. The pores of the reservoir are saturated with methane and methane hydrate. (ii) Consider three phases (gas phase, water phase, solid phase) and four components (free gas component, decomposition gas component, water component, hydrate component). Among them, the gas phase contains only methane gas, and the hydrate is treated as a solid phase; (iii) Only consider the two-phase flow of gas and liquid, and the fluid seepage conforms to Darcy's law; (iv) Homogeneous formation, i.e., porosity, permeability is constant; (v) Neglect the diffusion of gas and the dissolution of gas in water.”

Thanks for the careful review and constructive comments. We have accommodated your suggestions in the paper. We look forward to hearing from you regarding our submission. We would be glad to respond to any further questions and comments that you may have.

Reviewer 2 Report

This paper investigates the gas production characteristics in complex structure wells by numerical simulation.

The general approach may be feasible, and the discussion on the results obtained is detail. The goal of the study seems very clear. A few things need to be rectified before publication.

  1. provide a nomenclature part since too many symbols are used in the paper.
  2. provide an illustration of the problem that is studied in this paper.
  3. the mathematical models were provided in this paper , however the corresponding numerical methods and schemes are not mentioned, please provide the missing information about the numerical methods/schemes, boundary treatment, models, grids used.

Author Response

Response to Reviewer 2 Comments

Point 1: This paper investigates the gas production characteristics in complex structure wells by numerical simulation. The general approach may be feasible, and the discussion on the results obtained is detail. The goal of the study seems very clear. A few things need to be rectified before publication.

Response 1:

Thanks for your recognition of our paper work. We have carefully revised the manuscript.

Point 2: Provide a nomenclature part since too many symbols are used in the paper.

Response 2: 

Thanks for your comment. We have added a nomenclature part at the end of the paper:

Nomenclatures

The following nomenclatures are used in this manuscript:

A, B, C, q

constant

Ad

total surface area of the hydrate particles (m2)

Ahs, Adec

decomposition surface area (m2)

ch

hydrate concentration (gmol/m3)

Cr, Cg, Cw, Ch, Ci

specific heat of rock, gas, water, hydrate and ice (J/g/K)

E

the activation energy (J)

fe

fugacity of methane at the three phase equilibrium condition

fg

fugacity of methane in the gas phase

k

permeability (mD)

kd

hydrate decomposition rate constant

kod

intrinsic decomposition rate constant

kof

intrinsic formation rate constant

krg, krw, krh

relative permeability of each phase (mD)

ṁg, ṁh, ṁw

masses of gas, water, and hydrates decomposed per unit time (kg/s)

n, Nh

hydration number

Pe

hydrate three-phase equilibrium pressure (KPa)

Pg

gas phase pressure (KPa)

qg, qw, qh

injection/output quality per unit time and unit volume (kg/m3/s)

R

universal gas constant

Sg, Sw, Sh, Si

saturation of gas, water, hydrates and ice

t

time (s)

v

reaction speed (m/s)

xi, yi

mole fractions of methane in gas and liquid phase

ΔHh, ΔHi

heat absorbed/released per mole (J/mol)

λr, λg, λw, λh, λi

thermal conductivity of rock, gas, water, hydrate and ice (W/m/K)

μg, μw, μh

viscosity of each component (mPa s)

ρg, ρw, ρh, ρr

density of gas, water and hydrates and rock (kg/m3)

ϕ

porosity of the medium

Point 3: Provide an illustration of the problem that is studied in this paper.

Response 3: 

Thanks for your comment. We have improved the description of the problem description of this research in the introduction section:

“In addition, China’s first use of horizontal wells in the Shenhu waters of the South China Sea in 2020 proves that wells with complex structures are of great significance in the future research process of NGH industrialization, and it is not very clear about the improvement of NGH productivity of various complex structure wells, and there is no more complete combing. Therefore, this work evaluates the gas production characteristics of multi-branch vertical wells (MVW), multi-branch horizontal wells (MHW) and cluster horizontal wells (CHW) by numerical simulation, using the Nankai Trough formation in Japan as an example, with the aim of exploring the feasibility of complex structural wells and the status of capacity enhancement.”

Point 4: The mathematical models were provided in this paper, however the corresponding numerical methods and schemes are not mentioned, please provide the missing information about the numerical methods/schemes, boundary treatment, models, grids used.

Response 4: 

Thanks for your comment.

  1. For thenumerical methods/schemes, the following content has been added to Appendix A:

“The calculation and solution process is mainly as follows: (i) Enter the size of the simulation area, grid division, stratum parameters, boundary conditions and initial conditions; (ii) Import complex structure well point parameters and work arrangement; (iii) Calculate the decomposition rate of NGH; (iv) Calculate the saturation of each phase and the fraction of gas phase components; (v) Update the model parameters after each analysis step, and judge the control conditions for the end of the calculation. If the end time is not reached, then skip to process (ii) and continue the calculation.”

  1. For theboundary treatment, the following content has been added to section 2.2:

“The stratum is generalized as a horizontally extended stratum without considering the influence of the dip angle of the stratum on the flow of gas and water in both phases, and the upper and lower sides of the model are set as constant temperature and constant pressure boundaries where fluid migration and heat exchange can occur: (The system can't type the formula, please see the attachment. Thank you)

where n is the number of NGH; C is the index (the value in this paper is 1.57); q is a constant. The initial formation pressure field distribution of the model in this paper is determined by the change of the seafloor surface pressure of 10.72MPa according to the pressure gradient of 10kPa/m, and the temperature field distribution is determined by the seafloor surface temperature of 3.75°C according to the change of the geothermal gradient of 0.03 °C/m. Then a three-dimensional geological model is established(Fig 2).”

  1. For the description of the model, we have added Appendix A for further explanation.
  2. For the expression of grid usage information, we have improved it in section 2.2:

“To minimize the influence of boundary conditions, the model is expanded to 1000 m in X and Y directions; the total thickness in Z direction is 90 m, of which the total thickness of the NGH reservoir is 60 m; the thickness of overburden layer is 30 m; the thickness of the underlying layer is 30 m. The grids of the overlying layer and the underlying layer are argillaceous sediments (permeability less than 15mD) and sandy sediments (permeability greater than 15mD), with a size of 10m×10m×6m. The grid size of the NGH reservoir is 10m×10m×2m, and The specific reservoir parameters and other stratigraphic parameters are shown in Table 1. A total of 408040 grids (101 grids along the x, y-coordinate, and 40 grids along the z-coordinate) is used for the numerical simulations in this study.”

Thanks for the careful review and constructive comments. We have accommodated your suggestions in the paper. We look forward to hearing from you regarding our submission. We would be glad to respond to any further questions and comments that you may have.

Round 2

Reviewer 1 Report

The  authors responded to all the raised comments and remarks. The manuscript deserves to be published as a paper in journal Mathematics.